# Estimated Burden of Serious Fungal Infections in Ghana

**DOI:** 10.3390/jof5020038

**Published:** 2019-05-11

**Authors:** Bright K. Ocansey, George A. Pesewu, Francis S. Codjoe, Samuel Osei-Djarbeng, Patrick K. Feglo, David W. Denning

**Affiliations:** 1Laboratory Unit, New Hope Specialist Hospital, Aflao 00233, Ghana; obkatey91@gmail.com; 2Department of Medical Laboratory Sciences, School of Biomedical and Allied Health Sciences, College of Health Sciences, University of Ghana, P.O. Box KB-143, Korle-Bu, Accra 00233, Ghana; fscodjoe@chs.edu.gh; 3Department of Pharmaceutical Sciences, Faculty of Health Sciences, Kumasi Technical University, P.O. Box 854, Kumasi 00233, Ghana; osdjarb@yahoo.com; 4Department of Clinical Microbiology, School of Medical Sciences, Kwame Nkrumah University of Science and Technology, Kumasi 00233, Ghana; pfeglo@gmail.com; 5National Aspergillosis Centre, Wythenshawe Hospital and the University of Manchester, Manchester M23 9LT, UK; david.denning@manchester.ac.uk

**Keywords:** fungal infections, Ghana, HIV, TB, candidiasis, aspergillosis, tinea capitis

## Abstract

Fungal infections are increasingly becoming common and yet often neglected in developing countries. Information on the burden of these infections is important for improved patient outcomes. The burden of serious fungal infections in Ghana is unknown. We aimed to estimate this burden. Using local, regional, or global data and estimates of population and at-risk groups, deterministic modelling was employed to estimate national incidence or prevalence. Our study revealed that about 4% of Ghanaians suffer from serious fungal infections yearly, with over 35,000 affected by life-threatening invasive fungal infections. Incidence of cryptococcal meningitis, *Pneumocystis jirovecii* pneumonia, and disseminated histoplasmosis cases in AIDS was estimated at 6275, 12,610 and 724, respectively. Oral and esophageal candidiasis collectively affect 27,100 Ghanaians and 42,653 adult asthmatics are estimated to have fungal asthma. We estimate a prevalence of 12,620 cases of chronic pulmonary aspergillosis (CPA and an incidence of 1254 cases of invasive aspergillosis (IA). Estimated cases of candidemia and candida peritonitis cases were 1446 and 217, respectively. The estimated prevalence of recurrent vulvovaginal candidiasis (RVVC) and tinea capitis was 442,621 and 598,840, respectively. Mucormycosis and fungal keratitis each may affect 58 and 810 Ghanaians. These data highlight the urgent need for intensified awareness to improve diagnosis and management.

## 1. Introduction

Fungal infections, in total, are remarkably common in humans and even more common in plants. They range from mucocutaneous infections, which do not threaten life, but affect quality of life (mostly occurring in apparently healthy individuals) to invasive fungal infections (IFIs), and are life-threatening and usually occur in immunocompromised individuals [1,2,3]. Serious fungal infections (SFIs), including IFIs, significant chronic infections, and complicated mucocutaneous infections, have an impact on quality of life, such as recurrent vulvovaginal candidiasis (RVVC) and tinea capitis. Fungal infections generally affect six main categories of patients: Cancer, transplant and HIV/AIDS patients; critical care patients (premature babies, patients in intensive care units (ICU), patients after major surgery); patients with respiratory diseases (severe asthma, tuberculosis (TB), chronic obstructive pulmonary disease (COPD) and cystic fibrosis (CF)); injury (eye, burns, trauma, skin especially in tropics); and skin, hair, and nails infections and genitals (i.e., vulvovaginal and penile candidiasis (thrush), in normal people [4]. Fungal allergy also occurs in non-immuncompromised patients with poorly controlled asthma, fungal rhinosinusitis and occupational disease as the main manifestations.

Over the past decades, there has been a substantial increase in fungal infections worldwide [5,6,7]. It is estimated that over a billion people are affected across the world resulting in approximately 11.5 million life-threatening infections and ~1.5 million deaths annually [4]. In fact, deaths from the top 10 IFIs are comparable to those from TB [8] and many more than malaria [9]. The rising trend of IFIs is due to an increase in populations at risk (immunocompromised patients) resulting from advances in patient management procedures and drugs [10]. However, the HIV/AIDS pandemic, the continuing scourge of tuberculosis, COPD, asthma, the increasing incidence of cancers, and intensive care are the major drivers of fungal infections in both developed and developing countries globally [11,12,13,14,15]. New risks have emerged, including severe influenza admitted to ICU (19% incidence) [16], poor hospital hygiene [17], multidrug-resistant *Candida auris* [18], and numerous new immunomodulatory agents, such as infliximab and ibrutinib [19].

Unfortunately, many countries and health institutions turned a blind eye to the effect of fungal infections, and deaths resulting from these infections are usually overlooked. The recent advances in diagnostics, robust screening programs, and expanded access to inexpensive antifungal drugs provide an unprecedented opportunity to reduce the burdens of ill-health and death from fungal diseases [4]. In countries with developed health systems, fungal infections are diagnosed and treated, although many are still missed and only identified at autopsy [20,21,22]. Advanced knowledge, early suspicion, rapid diagnosis of fungal infections, and prompt initiation of effective antifungal therapy are the keys to improving outcomes as well as reducing unnecessary therapy.

Efforts have been stepped up globally to create awareness about SFIs. To make a case for improved attention comparable to that of other infectious diseases, it is imperative to highlight the burden and changing epidemiology of SFIs. In the last four years, over 50 studies were published which estimate the burden of these infections in different countries [23].

The burden of SFIs in Ghana is unknown with only few epidemiological data and case reports available. Most of these case reports involved postmortem diagnosis, implying a low index of suspicion and treatment [24,25,26,27,28,29,30,31,32,33,34,35,36]. As in other low and middle- income countries (LMIC), absence of diagnostic tools and antifungal drugs coupled with insufficient training of healthcare professionals (regarding fungal diseases) ensures that the mortality and morbidity of SFIs remain unacceptably high. Furthermore, insufficient interest has been shown by public health and research institutions in Ghana due to ignorance or lack of awareness. There are no surveillance programs on fungal diseases in Ghana.

The aim of this study is to estimate the burden of SFIs in Ghana using deterministic modelling developed by LIFE (www.LIFEworldwide.org) which was successfully deployed for other countries. The overall aim is to obtain a current status report and a numerical tool for expanded advocacy, increased research, improved laboratory diagnostic capacity, and advanced clinical expertise geared toward bettering the outcomes of patients with fungal disease while contributing to the global SFIs estimation project and awareness campaign.

## 2. Materials and Methods

A thorough search of publications in literature was undertaken to identify epidemiology papers reporting fungal infections rates in Ghana. The terms used included “fungal infections and Ghana,” “Candidiasis and Ghana,” “*Tinea capitis* and Ghana,” “Aspergillosis and Ghana,” “Cryptococcosis and Ghana,” “Histoplasmosis and Ghana,” and “Fungal keratitis and Ghana”. Where published data for the general population was not available, we used specific population at risk (HIV/AIDS patients, patients with pulmonary TB (PTB), COPD, asthma, cancer, and patients receiving critical care) and deterministic modelling to derive national incidence or prevalence estimates of SFIs.

General population demographics were obtained from the Ghana Statistical Service (GSS) official population projections based on the 2010 Housing and Population Census [37]. Information on specific populations at risk was sourced from local, regional and global databases and studies. They are (1) HIV/AIDS [38], (2) PTB [39], (3) COPD [40], (4) asthma (adults) [41], (5) hematological and lung cancer [42], and (6) critical care [43].

Cryptococcal meningitis (CM) was estimated in adults only (it rarely occurs in children) at a rate of 12.7% among adult HIV/AIDS patients with CD4 less than 200/μL [44]. *Pneumocystis* pneumonia (PCP) frequency was estimated by assuming 11% of newly diagnosed HIV/AIDS adults and 35% HIV/AIDS children are infected [45,46]. Histoplasmosis cases were estimated to occur at a rate of 1.5% over two years in HIV patients with CD4 counts <200 μL [47].

Invasive aspergillosis (IA), chronic pulmonary aspergillosis (CPA) and allergic bronchopulmonary aspergillosis (ABPA) and severe asthma with fungal sensitization (SAFS) were estimated in adults only. CPA prevalence was estimated using the previously described approach taken by Denning et al. [48], where the number of annual PTB cases with cavities (22%) was multiplied by the incidence of CPA in cavities (22%) and the number of PTB cases without cavities (78%) was multiplied by CPA incidence (2%). An estimation of a five-year prevalence of CPA was made, assuming a 15% annual mortality or surgical cure rate [49]. PTB, COPD, asthma, post-pneumothorax, and sarcoidosis can all trigger CPA. To calculate all cases of CPA, PTB was assumed to be the underlying disorder in 50% (range 20–80%) of cases [50]. ABPA estimation was made assuming 2.5% of adult asthmatics have ABPA [49]. Estimate of SAFS was obtained by assuming that of the worse 10% of adult asthmatics, 33% are sensitized to one or more fungi [51]. Invasive aspergillosis was estimated in hematological and lung malignancies, HIV/AIDS and COPD. It was assumed that 10% of acute myeloid leukemia (AML) patients develop IA and that an equal number of cases are found in non-AML hematological patients while 1.3% of admitted COPD patients [52,53], 2.6% of lung cancer patients [54] and 3.5 per 1000 HIV/AIDS patients [55] develop IA. Mucormycosis was estimated to occur at a rate of 0.2/100,000 (a general literature estimate) [56].

Candidemia cases were estimated assuming it occurs at a rate of five per 100,000 with 30% in ICU (critical care and post-surgical patients) and 70% in cancer and other immunocompromised patients [57]. For candida peritonitis (intra-abdominal candidiasis), we assumed that the rate was half of the ICU candidemia rate [58]. The estimated prevalence of RVVC was established assuming a frequency rate of 6% in adult women [59,60]. To estimate the number of cases of oral candidiasis (OC), we assumed that it occurs in 90% of HIV patients with CD4 counts <200/μL [61] and for that of esophageal candidiasis (EC), we assumed 20% of new AIDS cases and 0.5% of those on ART [62,63].

Cases of fungal keratitis were calculated using a study on suppurative keratitis undertaken in three hospitals (in three different regions) covering a population of nearly two million Ghanaians. In this two-year prospective study, 38% of the 290 suspected cases of suppurative keratitis were identified as fungal keratitis [33]. A crude rate of 2.8/100,000 was used. The estimate for tinea capitis was obtained from two studies which reported an average prevalence of 9% amongst schoolchildren in Ghana [24,34].

Annual incidence rates and prevalence were estimated per 100,000 inhabitants. Prevalence was calculated for CPA, ABPA, SAFS, recurrent VVC, and tinea capitis, the remainders of the estimates are annual incidence.

## 3. Results

Table 1 shows the estimates of total burden of serious fungal infections and the number of infections classified according to the major at-risk group as well as the rate per 100,000 inhabitants.

Ghana is a tropical country situated in the western part of Africa. Ghana is located between latitudes 4” and 11” N of the equator and shares common borders with Togo to the east, Burkina Faso to the north, Cote D’Ivoire to the west, and the south bounded by the Gulf of Guinea. In 2017, the Ghanaian population was projected to be 28.9 million, with 62% adults. The number of children between 5–14 years and females between 15–49 years was 6,653,773 and 7,377,009, respectively [37]. Ghana’s gross domestic product (GDP) per capita was $1642 per person in 2017, a major increase from $923 in 2006 [64] (Table 1).

In 2017, people living with HIV/AIDS (PLWHIV) was estimated at 310,000 with 125,667 on antiretroviral therapy (ART). Adults aged 15 and over living with HIV were 280,000. Nineteen thousand new infections and 16,000 AIDS deaths were also estimated [38]. There were 48,269 adult HIV/AIDS patients with a CD4 count of less than 200/μL and thus particularly at risk of serious fungal infections due to severe immunosuppression [65]. In 2017, the incidence of TB in Ghana was approximately 44,000 with 14,550 TB notified cases 92% of these are PTB [39]. There are an estimated 43,047 adult asthmatics in Ghana. Annual COPD hospital admissions (assuming 10% of COPD patients are admitted per year) were estimated at 16,143. GLOBOSCAN 2018 estimated a five-year prevalence of 3853 and 252 hematological and lung cancer cases respectively in Ghana. The estimated number of critical care beds in 2016 was 709 (assuming 10% of hospital beds are occupied by critically ill patients). Transplants programs (solid organ and hematopoietic stem cell) are currently not available in Ghana and there is no nationwide registry of surgeries.

Opportunistic fungal infections are major complications of HIV/AIDS. CM is a common opportunistic infection in HIV/AIDS patients. We estimated as many as 6275 CM cases. PCP is a life-threatening fungal infection complicating HIV/AIDS. It is known to be a leading initial presentation of AIDS, but is poorly described in Ghana. Estimated cases of this seemingly nonexistent infection in Ghana were 12,610 (90% of which occur in children). Oral candidiasis was calculated to affect 17,100 Ghanaians at a rate of 59 per 100,000 person-years, while esophageal candidiasis cases were 10,000. Based on numerical data from Tanzania, 724 cases of disseminated histoplasmosis were estimated to occur in AIDS patients annually.

Respiratory diseases including TB and asthma are major causes of morbidity and hospital admissions in Ghana. Sequelae of TB include development of *Aspergillus* antibodies indicating infection, usually CPA or bronchitis in those with post-tuberculous bronchiectasis. Exacerbations of asthma are common and may be driven by fungi, especially *Aspergillus* spp, which is common in the environment. ABPA, CPA, and SAFS frequencies are thus anticipated to be high. Eighteen thousand three-hundred eighty-five adults were estimated to have ABPA complicating asthma. Using previously described assumptions, estimated annual CPA incidence and five-year prevalence post-TB was 2002 and 6310, respectively [48]. Overall, we estimated 12,620 cases of CPA from all underlying diseases (ranges from 7888 to 31,550) at a prevalence of 44 cases per 100,000 person-years assuming 50% occurring post TB and the rest complicating other pulmonary conditions including COPD, asthma, post-pneumothorax, and others. We calculated that SAFS affects 24,268 Ghanaian adults annually. ABPA and SAFS are collectively known as “fungal asthma” which is responsive to antifungal therapy, and over 40,000 adults probably suffer from this. CF is probably rare in Ghana and absence of supporting data reporting CF in Ghana makes the estimation of *Aspergillus* infections in CF patients difficult. We estimated 1254 cases of IA in HIV/AIDS, hematological and lung cancer patients as well as those admitted to hospital with COPD occurring at a rate of four cases per 100,000 person-years.

An estimated 709 beds are assumed to be occupied by critically-ill patients in ICU including neonatal ICU (NICU). Using previously described assumptions, candidemia and *Candida* peritonitis was estimated to affect 1446 and 217 patients respectively per year, at rates of five and 0.75 per 100,000 person-years, respectively.

RVVC is defined as four or more episodes of vulvovaginal candidiasis per year. We estimated 442,621 RVVC to occur among adult women in the general healthy population at a rate of 1530 females per 100,000 person-years. We estimated that 598,840 schoolchildren suffer from tinea capitis at a rate of 2070 per 100,000 persons. Mucormycosis and fungal keratitis were estimated to occur in 58 and 810 Ghanaians respectively per year.

To the best of our knowledge, there are no reliable epidemiological data for the fungal Neglected Tropical Diseases (NTDs) mycetoma, chromoblastomycosis or sporotrichosis in Ghana. Although they are expected to occur among Ghanaians, they are probably rare.

## 4. Discussion

About 4% (1,147,228) Ghanaians were estimated to be affected by SFIs with 35,000 suffering from IFIs associated with high morbidity and mortality. This figure is more than the 3% reported in Tanzania [66], and much less than the 11.8% estimated in Nigeria [67]. Tinea capitis was the major contributor, constituting 52% of fungal infections. This is comparable to a previous study, which reported tinea capitis as the most diagnosed fungal infection in Ghana comprising 44% of all fungal diseases [68]. Unfortunately, the majority of these infections is not diagnosed or misdiagnosed which results in no treatment or wrong treatment. Our literature search showed only four case reports and two epidemiological data on IFIs in Ghana, with diagnosis made post-mortem in three of the reported cases [26,27,28,29,30,35].

Ghana had an estimated ART gap of 60% and AIDS death stood at 16,000 in 2017 [41]. There are probably many AIDS-related opportunistic fungal infections seen in this population as well as those on ART. AIDS deaths stood at 16,000 in 2017. Nearly half of AIDS deaths are reportedly caused by opportunistic fungal infections [4]. CM is reportedly responsible for ~15% of AIDS-related deaths [65]. Cryptococcal antigenemia (CRAG) is detectable a median of 22 days before the onset of symptoms [69] and was shown to be 100% sensitive for predicting the development of CM in the first year of ART [70], and is also associated with both CM and mortality [71]. In view of these observations, screening for subclinical or asymptomatic infection by a serum CRAG assay in patients with advanced HIV infection, and giving antifungal therapy to those testing positive, may prevent the development of CM. CM is infrequently diagnosed in patients admitted with advanced HIV infection in Ghana. The CRAG test is not yet routinely available for almost all patients admitted to hospital or attending HIV clinics in Ghana. We estimated 6275 cases of CM among adult HIV/AIDS patients, using a 12.7% rate from a study in Nigeria [44]. However, two hospital-based studies in Ghana [28,30] reported a surprisingly low prevalence of rate of 2% and they cannot be generalized to the population because of selection bias such that many patients with cryptococcal infection probably died before either being tested for HIV or attending the clinic. Our estimate is substantially higher than the estimate of Rajasingham et al. from 2017 of 636 cases and 540 deaths [65]. Accurate data on rates of cryptococcal antigenemia in patients admitted with advanced HIV infection or AIDS are required.

PCP occurs worldwide and is especially common in children. In the developed world, it is a common opportunistic infection in HIV-infected patients with previous reports suggesting it is a less important cause of morbidity in HIV patients in Africa. However, a recent study in Cameroon reported a high prevalence of *Pneumocystis jirovecii* detection among healthy HIV patients [72], while another also reported increased frequency of clinical disease in African children [73]. Additionally, a rising GDP is expected to increase PCP incidence [74], and Ghana had a massive increase in GDP over the last six years. However, as in other LMIC, it is poorly described in Ghana with no published data on PCP. Nevertheless, as per the guidelines for ART in Ghana, co-trimoxazole prophylaxis (which is very effective in preventing infection) is initiated in all HIV-positive subjects [75]. This, however, does not solve the problem because studies reported PCP in HIV/AIDS patients on ART and co-trimoxazole [72,74], and the prophylactic dose is much lower than the treatment dose, which is especially relevant to new presentations of patients with advanced HIV infection. Although PCP also occurs in non-HIV infected individuals who are immunocompromised, we excluded such cases in our estimate due to the dearth of supporting data.

Histoplasmosis in Africa markedly increased since the advent of the HIV/AIDS epidemic but is grossly under-recognized. Histoplasmosis is poorly described in Ghana with no epidemiological data. However, a study by Oladele et al. reports *Histoplasmosis capsulatum* var *capsulatum* (HCC) and *Histoplasmosis capsulatum* var *duboisii* (HCD) coexist in Ghana, with 12 cases of histoplasmosis reported in the last six decades (1952–2017). HIV-infected patients accounted for 92% (11) of the cases [76]. The clinical presentation of histoplasmosis may somewhat mimic TB, thus many cases are misdiagnosed or undiagnosed. In 2017, 35% of pulmonary TB cases were smear-negative and 21% of TB patients with known HIV-status are HIV-positive [39]. Most diagnoses of histoplasmosis are made post-mortem, as pre-mortem confirmatory tests are unavailable. Our estimate could be an underestimate with a 13% rate of histoplasmosis among HIV-positive patients with persistent fever and coughing associated with cutaneous lesions reported by Mandengue et al. in Cameroon [77]. In AIDS, 10–40% of patients present with skin lesions, which could facilitate rapid diagnosis if biopsied and examined histopathologically.

Oropharyngeal candidiasis (OPC) is the commonest fungal infection amongst HIV-positive patients worldwide. Oral candidiasis was reported as the third commonest clinical oral presentation in HIV positive patients in Ghana after melanosis and periodontal disease [78]. *Candida albicans* is the dominant causative organism and is largely susceptible to fluconazole, which is currently the sole treatment regimen available in Ghana. However, a study involving 267 HIV-infected patients with oropharyngeal candidiasis reported a shift in *Candida* species distribution profile with increasing non-*Candida albicans* species which are usually resistant to fluconazole [31].

TB is endemic in Ghana and a major health priority of the government. We estimate 12,620 cases of CPA, 50% occurring post TB, or possibly a mis-diagnosis of TB. Surprisingly, only one reported case of pulmonary aspergillosis has been published [27]. Indeed, the authors reported it was a late clinical diagnosis made in a patient with a presumptive diagnosis of smear-negative TB and treated with anti-TB and antibiotic drugs without clinical improvement. There was massive clinical improvement following initiation of itraconazole [27]. This re-emphasizes the current low index of suspicion and unavailability of pre-mortem diagnostic assays in the country, which are urgently needed. The key laboratory diagnostic test, *Aspergillus* IgG antibody detection (formerly precipitins testing), is not available in Ghana, and fungal culture of sputum is both insensitive for CPA and not routine because of issues with environmental contamination. A recent cross-sectional study from Nigeria supports the existence of many such cases [79].

The number of people with asthma is increasing in developing countries with most of the nearly 500,000 asthma-related deaths occurring in LMICs, including sub-Saharan Africa [80]. Adult patients with severe asthma are more prone to be colonized and allergic to airborne fungi (mainly *A. fumigatus*, *Alternaria*, and *Cladosporium* spp) leading to SAFS or more rarely to authentic ABPA. Fungal asthma is barely diagnosed in adult asthmatics in Ghana regardless of their association with more severe attacks and possibly asthma deaths. Skin prick testing for allergens is not done, although it is inexpensive and straightforward. Our estimate of the frequency of SAFS assumes a rate of fungal sensitization in severe asthma in adults of 33%, which is a conservative rate based on UK and other European data since African data are lacking [81].

SFIs are rarely diagnosed in cancer and critical care patients in Ghana with clinicians initiating empirical therapy rather than actively investigating for the causative organism due to lack of specialist mycology laboratories. Deaths from cancer soared to 15,089 in 2017, a figure expected to increase if attention is not given to IFIs complicating cancer. IFIs compromise cancer and reducing cancer mortality is unlikely without improving fungal disease diagnosis and management. There is no data to validate or modify our estimate of IA, candidemia and candida peritonitis cases. Nevertheless, retrospective review of blood cultures over a four-year period (2010–2013) from NICU and the infant unit of the Korle-Bu Teaching Hospital, which is Ghana’s premier referral hospital, indicate an increasing trend in candidemia from 0% in 2010 to 3.3% in 2013 [82], which is similar to the results of a similar study among cancer patients in the same hospital [83]. Antifungal treatment options in Ghana are limited to fluconazole, itraconazole, topical micanozole, topical nystatin, griseofulvin, and terbinafine. Once diagnostics improve, access to at least all generic antifungals will be necessary to improve outcomes. With the increasing emergence of non-*albicans Candida* and their resistance to fluconazole [31,36,84], current treatment guidelines require other potent antifungals such as amphotericin B and flucytosine.

Mucocutaneous fungal infections are very common often without specific underlying medical conditions, although they may be the harbinger of advanced HIV disease. They are not life-threatening, but they can greatly affect the quality of life of affected individuals. Serious mucocutaneous infections include RVVC, fungal keratitis, tinea capitis, mycetoma, chromoblastomycossis, and sporotrichosis. A study in 2012 reported the prevalence of vulvovaginal candidiasis among Ghanaian women was 21% in a gynecology clinic [36]. However, data on recurrent forms that do cause significant morbidity and discomfort are nonexistent.

Blinding suppurative keratitis is an extensive problem in Ghana (especially during the harvest period and windy seasons), with two studies reporting fungi as the main etiological agent [32,33]. In resource-constrained countries like Ghana, fungal keratitis morbidity is mainly influenced by difficulties in patient management because of a lack of diagnostic facilities and appropriate treatment. Agricultural work is thought to be commonly associated with suppurative keratitis. However, a study in Ghana showed it is significant in several occupations and affected mostly students/teachers and traders [32].

The prevalence of skin diseases, particularly tinea capitis among African schoolchildren, is high and can be associated with complications such as kerionor favus. Tinea capitis prevalence among schoolchildren in Ghana ranges between 8.4% and 8.7%, which is low as compared to other SSA countries [34]. Our estimate could probably be an underestimate because the 9% rate utilized was from a study in which tinea capitis was clinically diagnosed with no laboratory investigation. Even so, it dominates the burden of SFI and thus it is imperative to identify new strategies for proper care and prevention especially in rural communities.

There are no data on mycetoma for an accurate estimate to be made, although it is certainly present based on clinical experience, mostly in rural agricultural communities. There are no reports of chromoblastomycosis in Ghana, albeit there are some from Gabon as well as neighboring Nigeria and Cote D’Ivoire [85]. Sporotrichosis and entomophthoramycosis have also not been described in Ghana. Nonetheless, cases of sporotrichosis are described from South Africa [86,87,88,89,90], Sudan [91], Zimbabwe [92], and Nigeria [93]. In these studies, the majority of cases of sporotrichosis were among mine workers. Considering the present upswing and expansion of mining activities across Ghana, particularly in illegal mining where standard regulation policies, including safety protocols, are not duly adhered to. Therefore, may be cases of sporotrichosis going undiagnosed or misdiagnosed.

Low indices of suspicion among clinicians are just one facet of the present situation. Gaps in the availability of diagnostics services and antifungal drugs are the most critical elements. Some fungal diseases, such as oral candidiasis, can be diagnosed clinically. Conversely, most SFIs, especially life-threatening IFIs, require laboratory and in some cases radiological investigations for confirmation and separation from other similar presenting conditions. Currently in Ghana, only direct microscopy/histopathology and blood culture are offered but are not widely accessible and are only obtainable in tertiary and some secondary health facilities as well as some few private medical laboratories. Fungal culture is distant from clinical settings and reserved for research purposes only. Unfortunately, antigen and antibody tests are generally absent in clinical practice, yet indispensable in the diagnosis of IFIs being more rapid and more sensitive than histopathology and culture.

Compounding the lack of some key diagnostics, some of the most effective antifungal drugs, particularly the polyenes and echinocandins, are not available in the country [94]. Meanwhile, those available especially antifungals drugs, such as amphotericin B, itraconazole, griseofulvin, and terbinafine, for systemic use are not always readily available and often expensive for those who need them.

In Ghana, the hottest months are March and April, with temperatures ranging from 26–32 °C, and the coldest month is August, with temperatures ranging 23–27 °C [95]. There is a relatively long dry season from mid-October to March. The driest month in Ghana is usually January and has a maximum rainfall of 14 mm. The rainy season is from March to July and the highest rainfall is in June with an average of 210 mm. Ghana’s vegetation can be classified into three zones comprising coastal savannah, tropical rainforest in the central part, and the Guinea savannah in the north. The warm sun, the regular rains, and the fertile soils always gives rise to exuberant vegetation. Although, as a result of economic activities, most of the forest has been depleted in Ghana. However, are still remnants of deciduous and semideciduous forest that provides the inhabitants with both economic and medicinal avenues. The savannah, which is the main vegetation in the northern and certain parts of the south, is endowed with birds and other mammals. The housing sector in Ghana is highly unregulated, therefore, the majority of homeowners and landlords operate within no established frame work and set prices as they deem fit, which has contributed to the housing deficit in the country with an estimated housing deficit ranging between 1.7 and 2 million [96]. Ghana’s policy of liberalization was extended to the private sector participation in education. However, the country’s literacy rate is about 50%. As a result, the majority of the people turn to ito agriculture and other farming activities that expose these populations to infectious agents responsible for many microbial infections including fungal infections. Unfortunately, in Ghana, major agricultural and other farming activities are carried out at the village levels with poor or no health facilities. Therefore, the value of the estimated burden of the SFI’s provided in this study could be underestimated. Therefore, a thorough epidemiological studies coupled with other scientific investigations is warranted to support these claims in Ghana.

By comparison of the estimates of the burden of SFI’s in Ghana across other countries, reports indicate that global estimates of SFI’s found 500,000 cases of PCP in AIDS and non-AIDS people around the world [97]. CPA post TB was estimated as 4.3% and 9% among the Colombia and Malawian population respectively [98,99]. Also, in Malawi, 670,900 cases of tinea capitis (Figure 1) among schoolchildren were reported in 2018. In a similar related study, 21% of schoolchildren in Rwanda were estimated to have tinea capitis [34,99]. CPA, which commonly complicates pulmonary TB treatment, was found to affect 4.9% of the Uganda population [100]. Although attempts were been made by LIFE (www.LIFEworldwide.org) to estimate the SFI’s in most countries, comparison of the incidences of fungal infections is difficult, as epidemiological data is not available in most countries [98].

## 5. Conclusions

SFIs in Ghana are probably more common than expected, with 1,147,228 (4%) Ghanaians affected every year. Further epidemiological studies and surveillance program, as well as registers for at-risk groups, are urgently needed to confirm or adjust our estimates. This study provides the metrics to drive a change in the status quo. The government through its various health institutions and agencies need to place SFIs on their radar and establish policies incorporating strategies to improve the diagnosis and treatment of patients suffering from SFIs, particularly those associated with high debility and mortality. Increased clinical and public health awareness is essential. Clinicians must expand their knowledge of fungal infections and be equipped with the necessary clinical expertise in fungal disease diagnosis and management especially life threatening IFIs. Critically, improving laboratory facilities and specialized training of medical laboratory scientists and pathologists are crucial to improve patient outcomes and inform public health. Finally, conscious efforts must be made toward adopting the first WHO essential diagnostics list and also ensure all WHO-endorsed essential antifungal drugs are available and widely accessible in the country.

## Figures and Tables

**Figure 1 jof-05-00038-f001:**
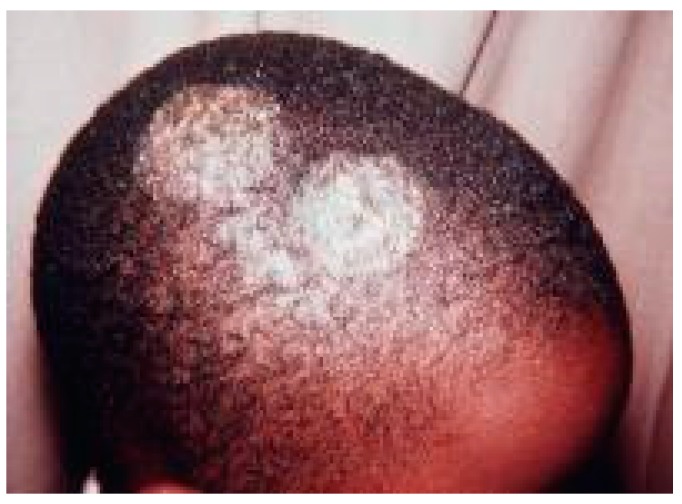
A patient presenting at the Kwame Nkrumah University of Science and Technology Hospital, Kumasi with fungal infection (tinea capitis) which is spreading to other parts on the scalp.

**Table 1 jof-05-00038-t001:** Estimate of Serious Fungal Infections in Ghana.

	No Underlying Disease	HIV/AIDS	Respiratory Disease	Cancer + Immunocompromised	Critical Care + Surgery		
Serious Fungal Infection						Totals	Rate/100,000
Cryptococcal meningitis	-	6275	-	-	-	6275	21.7
Pneumocystis pneumonia	-	12,610	-	-	-	12,610	43.6
Disseminated histoplasmosis	-	724	-	-	-	724	2.5
Invasive aspergillosis	-	977	210	67	-	1254	4.4
Mucormycosis	58	-	-	-	-	58	0.2
Chronic pulmonary aspergillosis	-	-	12,620	-	-	12,620	44
Allergic bronchopulmonary aspergillosis (ABPA)	-	-	18,385	-	-	18,385	64
Severe asthma with fungal sensitization (SAFS)	-	-	24,268	-	-	24,268	84
Candidemia	-	-	-	1013	433	1446	5.0
Candida peritonitis	-	-	-	-	217	217	0.8
Oral candidiasis	-	17,100	-	-	-	17,100	59
Esophageal candidiasis	-	10,000	-	-	-	10,000	35
Recurrent Candida vaginitis (RVVC)	442,621	-	-	-	-	442,621	1530
Fungal keratitis	810	-	-	-	-	810	2.8
Tinea capitis	598,840	-	-	-	-	598,840	2070
**Total serious fungal infection burden**						1,147,228

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
