# Peer review of "Estimated Burden of Serious Fungal Infections in Ghana"

_jof, 2019, doi:10.3390/jof5020038_

Round 1

Reviewer 1 Report

This study is one of the series of studies aiming to estimate the fungal burden worldwide. This study estimated the burden of SFIs in Ghana. One Interesting data is the estimated incidence of CM and PCP being really high, and the authors highlighted the importance of the PCP probably occurring mainly in children. I have only minor comments there are several misspellings/finger errors. Some of them are in

line 32 "totoare"

line 37 "patientsvis"

line 73 "hasbeen" "forother"

line 146 "infectionsare"

A my only suggestions is 

1. The authors compare the estimated proportion of people affected by SFIs with the estimated proportions in other countries, it'd be good to have IC95% to have a more objective comparison. 

Reviewer 2 Report

Please expand the Introduction (or Discussion) to include more detail about the climate, vegetation, housing, geographic distribution of population, and professions in Ghana. For example, manual agriculture puts farmers more at risk of environmental sources of cutaneous and subcutaneous mycoses.
